# Analysis and Countermeasures of China's Green Electric Power Development

Keke Wang [1,2], Dongxiao Niu [1,2], Min Yu [1,2], Yi Liang [3,4,*], Xiaolong Yang [1,2], Jing Wu [1] and Xiaomin Xu [1,2]

1  School of Economics and Management, North China Electric Power University, Changping District, Beijing 102206, China; wkk_ncepu@126.com (K.W.); ndx@ncepu.edu.cn (D.N.); ncepuym@163.com (M.Y.); yangxiaolong@neepu.edu.cn (X.Y.); 1182106009@ncepu.edu.cn (J.W.); xuxiaomin0701@126.com (X.X.)

2  Beijing Key Laboratory of New Energy Power and Low-Carbon Development, School of Economics and Management, North China Electric Power University, Changping District, Beijing 102206, China

3  School of Management, Hebei GEO University, Shijiazhuang 050031, China

4  Strategy and Management Base of Mineral Resources in Hebei Province, Hebei GEO University, Shijiazhuang 050031, China

*  Correspondence: louisliang@hgu.edu.cn

**Abstract:** The green development of electric power is a key measure to alleviate the shortage of energy supply, adjust the energy structure, reduce environmental pollution and improve energy efficiency. Firstly, the situation and challenges of China's power green development is analyzed. On this basis, the power green development models are categorized into two typical research objects, which are multi-energy synergy mode, represented by integrated energy systems, and multi-energy combination mode with clean energy participation. The key points of the green power development model with the consumption of new energy as the core are reviewed, and then China's exploration of the power green development system and the latest research results are reviewed. Finally, the key scientific issues facing China's power green development are summarized and put forward targeted countermeasures and suggestions.

**Keywords:** power industry; green development; new energy; multi-energy mode

## 1. Introduction

Energy is an essential material foundation for the human to achieve economic development and improve living standards [1,2], which is particularly important for modern economy [3]. In the face of growing energy shortages and environmental problems, the main strategies of most countries in the world are to develop and utilize renewable energy, promote the construction of a clean, low-carbon, safe, and efficient energy system dominated by renewable energy [4], adopt energy-saving and emission reduction strategies [5], and develop renewable energy technologies [6]. The existing energy system in China has provided the inexhaustible impetus for its economic rise [7]. According to OECD (Organization for economic cooperation and development) statistics; China has become the largest primary energy consumer and greenhouse gas emitter in the world. With the ever-increasing constraints on energy resources, the contradiction between extensive use of energy and ecological civilization's construction has become increasingly irrational [8]. Resource and environmental problems have become the key to restrict China's social development, so it is urgent to control greenhouse gas emissions [9]. Green and low-carbon development has become a significant strategy for China's economic and social development and a critical approach to ecological civilization construction [10]. The power industry is at the core position in the modern energy system. As a significant industry of energy consumption and pollutant emissions, it needs transformation and upgrading and plays a vital role in reducing greenhouse gas emissions [11]. Efforts should be made to develop green power [8] (refers to renewable energy power supply, which is easier to

obtain than traditional power [12], with wind power, solar energy, and biomass energy as the core). However, due to the randomness, volatility, intermittency, and uncertainty of renewable energy power generation, large-scale access brings severe challenges to the stable and safe operation of the power system [13,14], which restricts the green development of China's power industry. Therefore, it is urgent to study the weak links in the green development path of China's electric power, promote the transformation and upgrading of China's power green, and make remarkable contributions to building a beautiful China and promoting the construction of a community of shared future for humanity.

Under the background of large-scale development of new energy and power market reform, combined with the national macro policy, based on macro-historical data, this paper studies the green development path of Electric power in China to promote the transformation and upgrading of the power industry and better serve the society. Firstly, based on the national macro policy, this paper combs the macro development goals of China at this stage and analyzes the pressure of green power development in China. Secondly, based on macro historical data, this paper summarizes the achievements of green power development in China and explores the main problems existing at this stage. Then, this paper expounds the significance of various energy system models derived from the deepening of power system reform on the green development of electric power, focusing on the mechanism of multi-energy system participating in the market of green power; Finally, the paper analyzes the trend of power development in the future, and puts forward the countermeasures and suggestions to promote the green development of electric power in China.

## 2. Green Development Status of Electric Power in China

### 2.1. Brief Description of China's Electric Power

2.1.1. The Proportion of Clean Energy Consumption Is Increasing

During the "13th Five-Year Plan" period, China's energy consumption growth has gradually slowed down, and the overall consumption structure has been optimized. China is on the road to "building a clean, low-carbon, safe, and efficient energy system" and gradually promotes a clean and low-carbon consumption model.

2.1.2. The Green Transformation of the Power Supply Structure Continued

To prevent and resolve the risk of coal-fired power overcapacity and accelerate the green development of electric power, the growth rate of the new installed capacity of China has gradually slowed down, and the power structure has been continuously optimized. Wind power and photovoltaic power generation (PV) have gradually become an important part of renewable energy [15–17]. Since 2010, the ratio of installed thermal power in China has been gradually decreasing and the proportion of installed non-fossil energy has been increasing gradually. Installed wind power, photovoltaic, and other new energy power generation have been multiplying, becoming the new main force of power supply, and the installed structure of power supply has been constantly optimized. The form of China's installed generating capacity from 2010 to 2018 is shown in Figure 1 [18].

2.1.3. The Electricity Consumption Structure Presents a Clean and Low-Carbon Trend

As a clean and efficient final energy, electric energy will play an essential role in China's future energy structure. The proportion of electricity in the final energy consumption is a vital indicator to measure a country's final energy consumption structure and degree of electrification. Increasing the proportion of electric energy in final energy consumption is a vital measure to optimize China's energy consumption structure. In 2019, the electricity consumption of the whole society in China was 7225.5 billion kWh. In terms of industry types, the primary industry's electricity consumption is 78 billion kWh, and the electricity consumption of the secondary sector is 4936.2 billion kWh. The electricity consumption of the tertiary industry and residents' daily life is an essential driving force for the continuous and rapid growth of the whole society's electricity consumption, which will

still maintain double-digit growth in 2019. According to the trend of China's energy policy, it is the primary trend of future energy development to improve the level of final energy electrification. Electric energy substitution measures can be implemented to replace coal and oil with electricity to increase the proportion of electricity in final energy consumption.

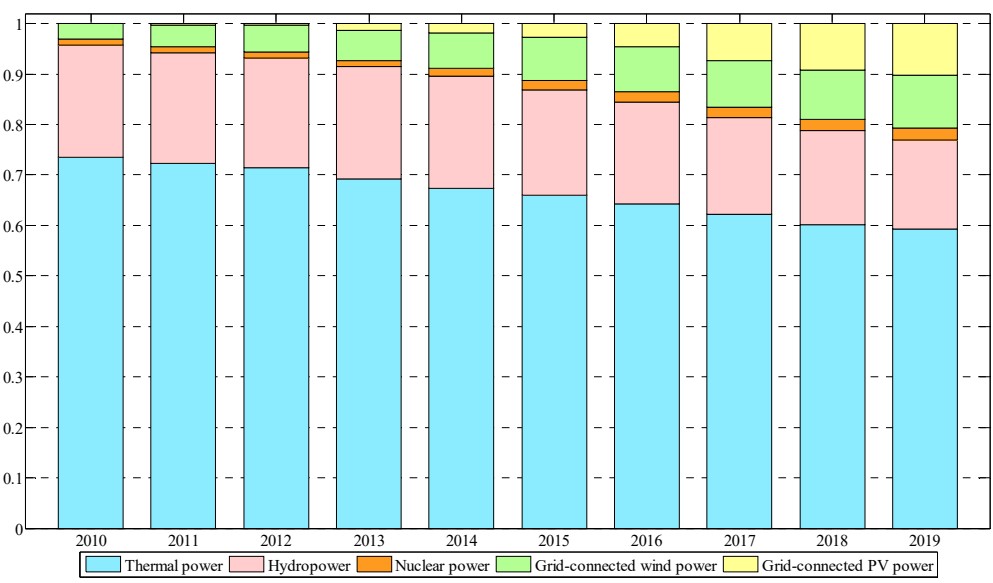

**Figure 1.** The installed power-generating capacity structure 2010–2019 in China.

### 2.1.4. Power Consumption and Environmental Indicators Are Gradually Optimized

The power industry actively responds to climate change and adopts various effective measures to reduce energy consumption and line loss rate [19]. In 2018, the national average standard coal consumption of thermal power plants of 6000 kilowatts and above was 307.6 g/kWh, and the coal consumption of coal-fired power generation units remained advanced in the world; the plant power consumption rate is 4.70% (including hydropower 0.26% and thermal power 5.81%); the national line loss rate is 6.21%; the hydro consumption per unit power generation of thermal power plants in China is 1.23 kg/kWh; the comprehensive utilization rate of fly ash and desulfurization gypsum is 71% and 74% respectively, and the total utilization capacity continues to increase. The emission of pollutants was further reduced. The emissions of soot, sulfur dioxide, and nitrogen oxides in China were about 210,000 tons, 990,000 tons, and 960,000 tons respectively, which were 19.2%, 17.5%, and 15.8% lower than that of the previous year; the emissions of soot, sulfur dioxide, and nitrogen oxides per kilowatt-hour of thermal power generation are about 0.04 g, 0.20 g, and 0.19 g; the waste water discharge per unit of thermal power generation is 0.06 kg/kWh. The carbon dioxide emission per unit of thermal power generation is about 841 g/kWh, a decrease of 19.4% compared to 2005, which has contributed to China's early realization of its carbon emission commitments.

### 2.2. Necessity of Green Power Development
### 2.2.1. The Embodiment of Active International Responsibility

As the largest developing country globally, China has pledged to reduce its carbon emission intensity by 40–45% by 2020 compared with 2005, and by 2030 it will reduce its carbon intensity by 60–65% compared with 2005. To realize its carbon emission commitment, China has made great efforts to develop an economic model with efficient use of resources and environment friendly as the core and the principle of "3R" (reduce, reuse, resources), actively promote energy substitution and optimize energy consumption structure to improve the increasingly severe environmental problems [20]; meanwhile, China will improve the policies of energy policy including energy tax policy, energy investment policy, industrial policy and energy consumption policy, and propose quota, and emission

trading plan. In December 2017, taking the power generation industry as a breakthrough point, the national carbon emission trading system was officially launched by the national development and Reform Commission. Other high energy consuming and high emission industries were gradually having to expand the market coverage [21]. This system is expected to include more than 3 billion tons of carbon emissions, surpassing the EU carbon emissions trading system, and becoming the world's largest carbon market [22]. Whether China can achieve the goal of carbon emission intensity by 2030 has been highly concerned. Relevant scholars make reasonable predictions based on China's national conditions, and the prediction of carbon emission intensity target is relatively ideal [23–25]. However, to achieve the goal of carbon emission intensity smoothly, China still needs to play the role of government supervision and policy guidance [26].

2.2.2. A Vital Measure to Achieve National Macro-Strategic Goals

The energy white paper published in 2007, "China's Energy Situation and Policy" started China's energy change [27]. At present, China's energy strategy is to promote the transformation of the energy development mode. By 2020, a "safe, green, and efficient" energy system should be initially constructed and formed by 2030. There are many macro development strategic plans in China have put forward explicit constraints on energy structure and consumption, as shown in Table 1. The implementation of the first energy law is accelerating, and renewable energy is about to become a priority area for energy development.

**Table 1.** National macroscopic strategy planning.

| China's Economy Has Entered a Period of New Normal | The Growth Rate of Primary Energy Demand Remains at a Single-Digit Percentage Level. |
| --- | --- |
| *The 13th Five-Year Plan for Energy Development, Strategic Action Plan for Energy Development (2014–2020)* | During the 13th Five-Year Plan period, the proportion of non-fossil energy power generation will reach 31%; during the 13th five-year plan and the medium and long term, the growth rate of China's energy consumption will further slowdown, and it will get the peak of energy consumption in 2040. |
| *Energy Production and Consumption Revolution Strategy (2016–2030)* | In 2020, the balance of non-fossil energy will reach 15%; from 2021 to 2030, the proportion of non-fossil energy consumption will reach20%; in 2050, the ratio of non-fossil energy will exceed 50% |
| *Medium and Long Term Development Plan of Renewable Energy (2007–2020)* | By 2020, the proportion of non-fossil energy in primary energy consumption will reach 15%. |
| *13th Five Year Plan for Renewable Energy Development* | By 2020 and 2030, the proportion of non-fossil energy in primary energy consumption will reach 15% and 20%. |

## 3. Challenges of Green Development of Electric Power in China

### 3.1. Resource Endowment Restricts the Space for Carbon Emission Reduction

Under the influence of the resource endowment characteristics of "rich coal, lack of oil, and less gas" in China, thermal power will still be a primary supporting power source for a long period and the rapid development of wind power and solar energy will not be enough to change this power generation pattern. After 2020, China's renewable energy will have market competitiveness [28,29], and after 2030, China's thermal power industry will gradually enter into a recession [30]. At present, China's power supply structure is dominated by thermal power generation. Coal accounts for a relatively large proportion of energy consumption [31], accounting for 57.7% [32]. The study confirms that increased energy consumption is the main cause of increased carbon emissions [33–35]. To realize the promise of carbon emission reduction in 2030, it is urgent to change the current coal-based energy consumption structure through the green transformation of electricity.

### 3.2. Energy Efficiency Still Needs to Be Improved

During the "13th five-year plan" period, China implemented the "double control" action of total energy consumption and intensity, requiring that the energy consumption per unit GDP by 2020 be reduced by 15% compared with that in 2015, and the total energy consumption should be controlled within 5 billion tons of standard coal. In 2018, China's energy consumption per unit GDP was 2.41 tons of oil equivalent/10,000 US dollars [31]. Compared with the average world and developed countries, China's energy consumption intensity is still on the high side, as shown in Figure 2. Moreover, the primary energy consumption is highly dependent on coal, and there is much room for improvement. Energy efficiency has become a significant challenge for the green transformation of electric power.

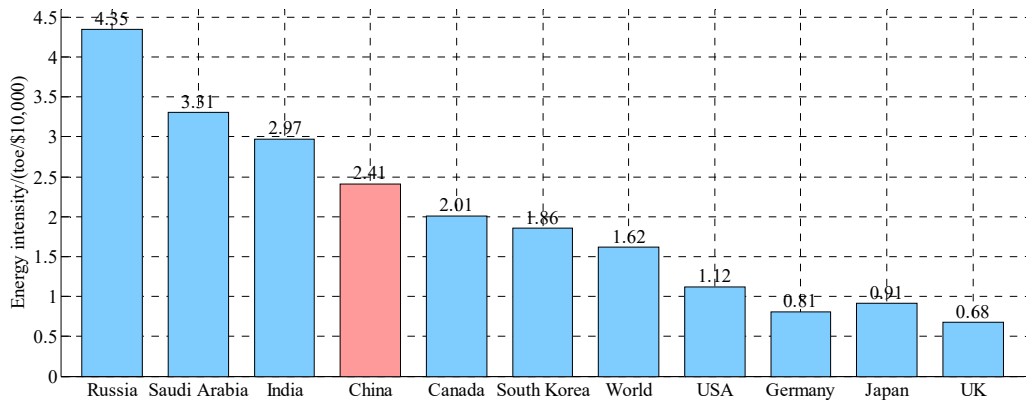

**Figure 2.** Energy consumption per unit GDP of major countries in the world in 2018.

### 3.3. The Contradiction between the Development and Utilization of New Energy

In 2019, the national average abandonment rate of wind and solar was 4% and 2%, respectively, and the phenomenon of abandoning wind and solar still exists. By 2020, China aims to install 350 million kilowatt-hours of hydropower, 200 million kilowatt-hours of wind power, and 110 million kilowatt-hours of solar power, respectively. It is urgent to solve the contradiction between the development and consumption of new energy. Otherwise, the phenomenon of energy abandonment will be further aggravated. For many years, to promote the development of the renewable energy industry, China has been adopting the mode of "benchmarking electricity price + government financial subsidy." The subsidy fund comes from the renewable energy price surcharge, which is charged with the electricity fee. With the rapid growth of wind power and photovoltaic installed capacity, the subsidy gap continues to expand. The existing pricing mechanism of new energy products has defects and the evaluation mechanism is not perfect, which is not conducive to the balanced development of the new energy industry [36]. The "*Notice on Matters Related to Photovoltaic Power Generation in 2018*" (Fagai Energy [2018] No. 823) stated that the principle of "reduction of both ordinary photovoltaic and distributed photovoltaic" was determined in the electricity price, and the scale of photovoltaic construction was reduced. The decline of new energy subsidies has brought great difficulties and uncertainties to the development of wind power and photovoltaic industries. The reduction of new energy subsidies has brought more significant problems and tension to the development of wind power and photovoltaic industries.

The distribution of renewable energy such as hydro energy, wind energy, and solar energy is highly concentrated in China and the resource-rich areas are in reverse distribution with the load centers; 60% of China's hydro energy resources are concentrated in the southwest, and the areas with better development conditions such as wind power and photovoltaic power are concentrated in the West and North. However, the eastern coastal areas, where the power load accounts for more than 2/3 of the total load in China, are economically developed, but energy resources are relatively scarce. It is difficult to flexibly

dispatch thermal power generation [37]. There is a lack of gas turbine power station with fast-tracking load adjustment and pumped storage power station (PSPS) with flexible peak regulation and frequency modulation in resource-rich areas. The overall accommodation conditions are insufficient, which restricts the utilization of renewable energy.

## 4. The Exploration of the Green Development Mode of Electric Power in China

To cope with the problems of the rapid growth of energy demand, environmental degradation, and low efficiency of energy conversion, countries in the world continuously adjust their energy structure and seek new ways of energy development. At present, the solutions are mainly divided into two categories: (a) clean energy gradually replaces fossil energy, enhances the dominant position of new energy and takes the development path of low-carbon and environmental protection; (b) adopting the ladder or recycling energy mode to improve energy utilization efficiency and build an integrated operation system of social energy [38]. China also proposed the "Internet+" Smart Energy Action Plan [39], which takes the power system as the core and uses the conversion capabilities between different energy sources to achieve synergy and complementarity of energy sources [40]. Multi-energy synergies and complementarities can accurately reflect the complex coupling relationships among energy networks [41,42], and the load types are diverse [43]. In this way, multiple energy sources can be complementary and mutually supported, and efficiently utilized [44], providing possibilities for the sustainable development of human society [45]. In this paper, the innovation mode of promoting the green development of electric power is divided into two types, namely, the multi-energy synergy mode and the multi-energy combination mode, and the promotion effect and positive significance of them to the green development of electric power are expounded.

### 4.1. Multi-Energy Synergy Model

With the continuous advancement of China's power system reform, various multi-energy system models have been gradually derived, namely, three typical operating models such as micro-grid (MG), integrated energy system, and internet of energy (IOE) [46]. The "multi-energy complimentary integration optimization demonstration project" was launched in 2017 [47].

### 4.1.1. The Proportion of Clean Energy Consumption Is Increasing Year by Year

In 2001, Professor R.H. Lasseter put forward the micro-grid energy system, which effectively solved the volatility and randomness brought about by a distributed energy grid-connected system. MG is essentially a distribution method. It is a small power generation and distribution system with an energy storage (ES) system, energy conversion device, related load, monitoring, and protection device, which is a significant component of the current smart grid [48,49]. MG can realize the integrated utilization of clean power such as wind and solar energy, coordinate user and energy transmission, promote local energy consumption, and improve energy utilization efficiency, leverage flexibility, and service level [50,51]. It is an effective means to effectively solve a series of problems caused by the direct connection of distributed generation to the grid [52].

Limited by the consumption capacity of renewable energy and the ability to cope with the renewable energy volatility [53,54], a single micro-grid is highly dependent on controllable power sources and ES devices, which limits its large-scale promotion and application [55]. Relevant scholars have proposed the coordinated interconnection mode of a multi-micro-grid (MMG) [56,57]. The MMG is a small power system integrated with power generation, distribution, and utilization [58], which can effectively solve the problem of power grid load growth, provide auxiliary services for the distribution network [59], and reduce the impact of renewable energy on the grid. Simultaneously, the MMG can be used as a flexible schedulable load to realize peak load shifting and valley filling [60].

### 4.1.2. Integrated Energy System

The integrated energy system is a kind of energy supply mode based on the gradual development of micro-grid technology, which breaks through the barrier of low utilization efficiency of traditional energy structure and realizes the coordinated planning and dispatching of electricity, gas, heat, cold, and other energy [61]. The integrated energy system is a new regional energy supply system characterized by improved energy utilization efficiency, full use of renewable energy, and collaborative and efficient utilization of conventional energy and renewable energy [62]. It is also an essential physical carrier of the IOE [63,64]. At this stage, no less than 70 countries in the world have conducted research on the relevant technologies of the integrated energy system to promote the sustainable supply of energy in all countries in the future [65]. Compared with the traditional energy system, the integrated energy system effectively realizes the interconnection and development of various energy equipment at the Internet layer, and meets the energy demand of multiple users. The comprehensive energy system is the core direction of future energy development transformation and transformation [66], a significant breakthrough point of China's energy revolution, and a powerful driver for China's construction of a "clean, low-carbon, safe, and efficient" modern energy system. Accordingly, it is of great practical and strategic significance to explore and build a theoretical system of comprehensive energy system suitable for China based on the actual needs of China.

Integrated energy systems will help address energy security issues, improve social efficiency, and promote the development of intermittent new and renewable sources of energy. Simultaneously, with the deepening of power market reform, an integrated energy system can integrate the energy side and demand-side resources and directly participate in power market transactions [67–69].

### 4.1.3. Internet of Energy

Energy industry is gradually decarbonization and digital transformation [70]. The emergence of IOE is based on clean and low-carbon energy, sustainable development and the growth of Internet, and information technology. It is the excellent integration of new energy technology and Internetworking technologies, and has the characteristics of multi-functional complementary, real-time two-way communication [71], open interconnection, peer-to-peer sharing, user-centered, and distributed [72]. IOE is an important technology to deal with today's energy crisis [73]. Scholars have not yet unified the definition of the concept of IOE. References [74,75] think that IOE is a large-scale utilization of distributed energy, which is a comprehensive energy system with power as the core and coupled with natural gas, electrified transportation, and other system networks; References [62,76,77] think that the IOE is proposed based on the new generation of intelligent utility network, which can realize the innovation of energy management mode; Reference [78] points out that IOE is an energy network with a large number of distributed, less centralized, and intensive consumer interaction; Reference [79] focuses on the transmission and use of multiple energy sources based on the power grid.

The critical significance of the IOE can be summarized as three points: (a) To realize multi-type energy interconnection and resource integration, effectively improve energy utilization efficiency [80], promote the transformation of energy utilization mode from extensive to intensive [81], realize large-scale clean energy substitution [82], and learn the optimal allocation of resources; (b) build a convenient energy supply and consumption platform, closely connect energy data information with user demand, maximize the use of new energy [83], and effectively promote new energy consumption [84], and achieve significant long-term emission reduction effect [85]; (c) using IOE is a critical way to solve the problem of high energy consumption and severe pollution in China [86], to alleviate the shortage of energy resources [87], and to build a clean, low-carbon, safe, and efficient energy system. Although the concept of the IOE does not have a uniform standard definition, the core value of the IOE has been widely recognized by scholars.

### 4.2. Multi-Energy Combination Model

Green development of electric power, clean energy has gradually replaced disposable energy, and its status has been improved continuously. With the continuous expansion of the grid connection scale of new energy, the intermittency, fluctuation, and reverse peak regulation characteristics of new energy generation power, as well as the reverse distribution of resources and load centers in China and other reasons, the phenomenon of "wind abandoning and solar energy abandoning" occurs [88–90], and the uncertainty of power system operation increases significantly [91]. Under the background that China is still dominated by thermal power generation, it has become an urgent problem to build a combination mode of thermal power and new energy, break the barriers between thermal power and new energy, encourage thermal power to play its role in the maximum extent, promote the consumption of new energy, and deepen the degree of green power in China. Scholars have carried out a large number of meaningful studies, in addition to three typical multi-energy collaborative modes, the combined model of thermal power generation, multiple types of clean energy and ES is adopted according to local conditions to improve the economy of power system and reduce the consumption of fossil energy [92,93].

One of the critical paths for the green development of electric power is to reduce the fossil energy use and use more readily available renewable energy to make it occupy a dominant position in the energy structure and energy consumption. However, the output of new energy mainly based on wind power and PV has a strong dependence on weather conditions, showing strong randomness. At the same time, the peak regulation frequency and amplitude of power systems are increased, which will increase the life cycle cost of thermal power units [94]. To solve this contradiction, some studies have pointed out that the application of ES technology can optimize the power supply structure [95] and improve power quality [96,97], the rapid charging and discharging characteristics of ES can improve the flexibility of the unit, eliminate the uncertainty of some new energy output, smooth the load curve [93,98], and improve the adjustable controllability of new energy generation [99], which provides a new solution for large-scale grid connection of new energy. Several studies have confirmed the positive effect of ES configuration on calming the power fluctuations of new energy [100–102]. At present, with the increasing maturity of ES technology, it can become a critical means of peak load regulation and to realize peak shaving and valley filling [102,103]. As a flexible and adjustable power supply, ES is not limited to conventional ES devices such as battery power station. Adjustable hydropower can also be regarded as a form of ES. The pumped hydro energy storage (PHES) units have a low construction cost and has been widely applied due to its profitability, ES capacity, and superior scheduling flexibility. The corresponding combination mode can reduce the intermittent influence of alternative energy output on the grid [104,105] and improve system operation efficiency [106,107].

Based on the complementary characteristics of multi-energy sources, some researches put forward corresponding multi-energy combination modes. References [108–110] confirmed the value of ES for renewable energy system with strong volatility. Reference [111] pointed out that ES can improve the economy and reliability of wind-PV combination system, and put forward the optimization strategy of ES capacity allocation. Reference [112] considered that the wind-solar-ES (PHES) combined system can effectively reduce the energy costs and make it possible to continuously supply power in remote areas. Reference [113] combines three different types of intermittent resources, namely wind, solar, and tidal energy with energy storage. The energy combination mode can smooth the power load disturbance and assist in the formulation of a reliable power generation plan. New energy with strong output volatility is equipped with energy storage, which can effectively promote the generation and utilization of new energy [114,115]. Reference [116] pointed out that in the future, a peak shaving and frequency modulation mode for combined operation of PHES or independent large hydropower stations can be established, which can give full play to the flexibility advantages of cascade hydropower stations in peak regulation and frequency regulation, relieve the pressure of thermal power peak regulation [117], and

improve the economy of power system operation [118,119]. On this basis, to make full use of the complementary characteristics of different energy sources, complex and diversified energy combination modes, such as wind-solar-thermal-storage, wind-solar-hybrid-gas-thermal-storage, etc., were proposed [120–124]. These combined models can give full play to the complementary characteristics of power supply in vast space and time, improve the flexibility of power system, and enhance renewable energy consumption. In addition to wind and solar, nuclear power will become a clean and reliable power generation form that can replace fossil energy on a large scale in the future, and its proportion in the power grid will gradually increase. Due to its inherent characteristics, nuclear power units are not suitable to undertake the task of peak regulation alone, so combining operation with units with fast response speed can better meet the peak regulation requirements of the power system [125]. To improve the safety and flexibility of nuclear power operation, we can choose nuclear—thermal, nuclear storage combined peak shaving strategy [98], nuclear thermal pumped storage combined peak shaving strategy, and nuclear thermal virtual power plant joint peak shaving strategy [126–128], and increase the peak shaving strategy of virtual power plant, which can effectively suppress the equivalent load prediction error and reduce the generation cost and carbon emission of the system. The multi-energy combination model proposed in the above research is shown in Table 2.

Under the background of green transformation and upgrading of China's electric power, facing many problems such as shortage of energy resources and consumption of renewable energy, the development of diversified energy mode is conducive to breaking through a power system's bottleneck itself. From the perspective of time and space, the complementary characteristics of energy are fully utilized to give new energy a wider space for consumption. At present, there is still a contradiction between the high concentration of new energy and the transmission of high-quality electric power resources. Large-scale wind power and photovoltaic power generation areas have small electricity demand and low power grid scale and grade. In the future, it is urgent to build gas turbine power stations that can quickly track load adjustment and PHESs that can flexibly adjust peak and frequency, so as to improve the overall absorption conditions of new energy, deeply tap the potential of multi-energy synergy mode, and promote the deepening of power green.

**Table 2.** Comparison of multi-energy combination modes.

| Reference | Combination Modes | Research Content | Objective Functions | Conclusions and Suggestions |
|---|---|---|---|---|
| [114] | Wind-energy storage (ES) (battery) | Energy storage assists peak shaving of thermal power units | The three-level optimization objectives from top to bottom are as follows: maximize the effect of peak shaving and valley filling and the operation economy, minimize the total peak shaving cost, and maximize the benefit of thermal power unit. | With the participation of energy storage system, the peak shaving capacity of the system can be improved, the wind abandonment can be reduced, the maximum peak valley difference regulation of thermal power units can be enhanced, and the total peak shaving cost of the system can be reduced at the same time. |
| [115] | Wind-ES (pumped hydro energy storage (PHES)) | Wind power-pumped storage combined daily operation to declare the next day output plan optimization | Maximize benefit of combined operation. | Wind storage combined operation can realize peak load shifting and valley filling, track load changes, and improve the overall operation economy. Increasing the start-up and shutdown times of PHES units can further improve the economic benefits and load tracking characteristics of wind storage combined mode. |
| [117] | Hydro-thermal-wind | Optimization model of combined operation of hydro- thermal-wind based on hydro and wind compensation principle | Minimize the fluctuation of thermal power generation and output, and minimize the total carbon emission. | The joint operation of hydro-thermal-wind can reduce the fluctuation of thermal power generation and the fluctuation of wind power output, effectively reduce the carbon emission of the power grid, improve wind power generation, and alleviate the phenomenon of wind abandon. |
| [118] | Hydro-wind-thermal | Optimal hourly generation plan of hydro-wind-thermal combined system | Minimize the fuel cost of thermal power plant total cost of hydro-wind-thermal joint operation, maximize the utilization of wind power and hydropower. | The hydro-wind-thermal combined system can effectively reduce the dispatching cost and promote the utilization of hydropower and wind power. |
| [119] | Hydro-wind-PV | Short term stochastic peak shaving optimization model of hydro-wind-PV system under multiple uncertainties | Minimize the peak-valley difference of load time series. | The combined hydro-wind-PV mode can effectively smooth the load and is beneficial to the hydro storage of the reservoir in the valley period. |
| [120] | Hydro-thermal-wind-PV | The complementary operation model of hydro-thermal-wind-PV connected power generation system suitable for day-ahead dispatching of power grid; the adaptive peak-regulating strategy for hydropower stations | Maximize the utilization of new energy for power generation and minimum carbon emission. | The hydro-thermal-wind-PV joint operation model can increase new energy generation, reduce thermal power generation and wind and PV, new energy output fluctuations, and alleviate the plight of new energy curtailment. |

**Table 2.** *Cont.*

| Reference | Combination Modes | Research Content | Objective Functions | Conclusions and Suggestions |
|---|---|---|---|---|
| [121] | Hydro-thermal-wind-ES (PHES) | Optimal generation scheduling of hydro-thermal-wind-es | Minimize the total cost of power generation in the combined system. | Pumped storage and wind turbines reduce the power generation cost of the system, and encourage the utilization of wind and hydropower in the multi-energy combination mode. |
| [122] | Hydro-thermal-wind-ES (Electric Vehicle) | Hydro-thermal-wind dynamic optimal scheduling for electric vehicles; the significance of large-scale electric vehicles for wind power grid connection | Minimize the power generation cost of the combined system; minimize carbon emissions; maximize the utilization of wind power. | The combination between electric vehicles and wind energy has the advantage of intelligent dispatch. Its participation in the joint system can increase wind energy utilization, reduce power generation costs, and have a positive impact on reducing carbon emissions. |
| [123] | PV-wind-hydro-thermal-ES (PHES) | Multi-region dynamic economic dispatch | Minimal total load cost in all areas. | The PV-wind-hydro-thermal combined system equipped with pumped storage effectively reduces the fuel cost of thermal power plants and the operation and maintenance costs of new energy power plants. |
| [124] | Wind-PV-hydro-gas-thermal-storage | Combined dispatching optimization of multiple power supply can give full play to the complementary characteristics of power sources and enhance the flexibility of power system | Multi-objective optimization with minimum total power generation cost and maximum renewable energy efficiency. | Compared with the traditional dispatching mode of "determining power by heat" and "determining power by hydro" for hydropower units, it is verified that the multiple joint dispatching mode has strong applicability in western provinces of China, which can improve the flexibility of system operation, improve the consumption energy of renewable energy of wind and PV energy, and enhance the overall operation economy of the system. |
| [126] | Nuclear-ES (PHES) | Nuclear power and pumped storage units combined participate in daily peak load regulation of power grid. | Minimum capacity of pumped storage power station (PSPS) with combined operation of nuclear power units. | The combined operation mode of nuclear and storage maximizes the peak-regulating performance of pumped storage and improves the daily generating capacity of nuclear power units, which is more economical. In the future, with the introduction of peak-valley electricity price, the combined operation mode of nuclear and storage will be more economical. |

**Table 2.** *Cont.*

| Reference | Combination Modes | Research Content | Objective Functions | Conclusions and Suggestions |
|---|---|---|---|---|
| [127] | Nuclear-thermal-virtual power plant | Three-stages combined peak shaving model of nuclear thermal virtual power plant based on combination of peak shaving resources on generation side and demand side | The carbon trading mechanism is introduced to analyze the operating cost of the system from two aspects of economy and low carbon, and the optimization goal is to minimize the combined peak-adjusting cost of the system. | The combined nuclear-thermal mode can determine the peak adjustment mode and peak adjustment depth of nuclear power units according to the equivalent load demand of the system, to relieve the peak adjustment pressure of thermal power units, reduce the start-stop peak adjustment of thermal power units, and reduce the operating cost of the system. Considering carbon trading cost, the nuclear-thermal-virtual power plant combination model can reduce the system's power generation cost and carbon emissions and realize the coordination and balance of economic and environmental benefits. |
| [128] | Hydro-Thermal-Wind-PV-Nuclear | Day-ahead unit power generation commitment plan for the hydro-thermal-wind-PV- nuclear combined system; a joint operation model of nuclear power units participating in peak shaving | Minimal total operating cost, peak shaving cost, and overflow loss cost of the combined system. | Make full use of the complementary characteristics of multiple power types to efficiently coordinate the operation of various power sources; improve the flexibility of unit operation, and increase the potential of system peak regulation; reduce the operating cost of thermal power units, the losses of hydro spillages and peak regulation costs. |

## 5. Exploration on Green Development System of Electric Power in China

The power industry is the critical field of energy conservation and emission reduction in China. To strengthen renewable energy development and promote its complete consumption is the core of the green development of electric power [129]. Facing the new wave of electricity marketization reform, designing a scientific and reasonable renewable energy consumption mechanism is the critical problem urgently solved in the current electric power industry.

To improve the increasingly severe environmental problems, accelerate the progress of green power development, and realize the carbon emission commitments in the Paris Agreement, China uses "visible hands" and "invisible hands" to improve increasingly severe ecological and environmental problems. Measures include actively increasing the percentage of cleaner alternative energy sources, speeding up the promotion of changing coal to coal gas, changing coal to electricity, and optimizing the energy consumption structure; besides, China has also improved energy policies, promoted green power trading mechanisms, established carbon trading pilots and power spot market pilots, etc. [130].

China implemented its Renewable Energy Law in 2006, but the share of non-hydro renewable energy in electricity production remains low. Compared with raw power, green power has no price advantage [11], and the reverse distribution of green power resources and power demand restricts the development of green power in China. Due to the bad influence of market mechanism regulation and public will on green power development, the support mechanism of green power should be established to adapt to the large-scale development of green power [1]. The green power products derived from this mechanism have become the means to promote the development of renewable energy. They are used to create and guide consumers' demand for green power [13].

As a critical area of energy conservation and emission reduction, the power industry has experimented with various green trading mechanisms, such as power generation rights, emission rights, carbon emission rights, and renewable energy quota system, and achieved absolute results [131]. Power generation right transaction is a unique alternative power generation transaction in China, a financial transaction between a generator set and a power plant to replace the production of contract power [132]. Power rights trading can be closed based on reasonable compensation for small units with high energy consumption and low efficiency. High-efficiency and large-capacity units can obtain more generation indexes simultaneously. This green trading mechanism can optimize the power industrial structure [133] and realize energy conservation and emission reduction. Emission trading is a pollution control method to achieve the goal of pollutant reduction by providing economic incentives [134]. Currently, the most successful market is the European Union's carbon emission trading market. Since 2002, China has launched a pilot trading scheme for sulfur dioxide emissions and carbon emissions in 2013. China has also launched a pilot trading scheme for emissions of pollutants such as nitrogen oxide and soot. Studies have shown that conducting emissions trading can reduce the unit emission reduction costs of pollutant emissions [135] and enhance the competitiveness of renewable energy sources [104]. In December 2017, taking the power generation industry as a breakthrough point, the national carbon emission trading system was officially launched by the national development and Reform Commission, other high energy consuming and high emission industries were gradually included to expand the market coverage [136]. As a policy tool to implement renewable energy quota system, the green certificate can promote the consumption of a renewable energy and the development of non-hydro power renewable energy through marketization [137].

In the power industry, a variety of green trading mechanisms is carried out to save energy and reduce emissions and promote the green development of the industry. However, due to the mutual influence of various green trading mechanisms and their own business, the trading contents are also different. There is no linkage mechanism in the initial allocation, pricing method, management mechanism, and so on. Power enterprises cannot form a benign overall planning strategy and ultimately cannot play a critical role in resources. The

green trading mechanism is the compensation for the ecological environment of enterprises in the process of energy utilization. It is the ecological cost of energy consumption of enterprises. If a variety of green transaction mechanisms are parallel, there will be repeated payment of environmental expenses and little cost-sharing [138], which is not conducive to giving full play to the enthusiasm of enterprises in energy conservation and emission reduction, and the implementation of green trading mechanism is increasingly difficult.

Existing literature has carried out research on different green trading mechanisms. Based on the characteristics of green trading tools, corresponding portfolio trading patterns have been proposed to realize the linkage of various trading mechanisms. Reference [139] mentions that carbon emissions trading will enhance the market competitiveness of renewable energy. However, the relevant renewable energy support plans and carbon trading mechanisms overlap each other, and it is necessary to analyze the contradictions and synergies of different mechanisms. Studies have shown that running the renewable energy support program and the carbon trading market at the same time can achieve sustainable carbon emissions reduction and optimize the future energy structure [140]. Reference [141] takes the maximization of the net present value of power generation expansion investment as the optimization goal. The analysis shows that green certificates and emissions trading may be powerful tools to promote renewable energy development. Reference [142] simulates the joint dynamic changes of the electricity market and the green certificate trading market and considers the interaction between the price two markets' price shocks. It is confirmed that the green certificate market will fully stimulate the increase of the installed capacity of green energy represented by wind power, thus reducing the benefits of wind power in the power market. Based on the theoretical analysis of "rights" and "shares," reference [143] considers that there is a strong correlation between generation rights trading and emission trading in terms of mechanism design, so it is suggested to conduct combined trading. References [144,145] considers that generation rights replacement with carbon trading can effectively reduce carbon dioxide emissions and improve the power producers' enthusiasm. Reference [146] studies the impact of initial carbon emission share allocation on generation right trading, and it considers that the carbon emission right allocation mode inclined to high-efficiency units to a certain extent is more conducive to the development of generation right trading. In reference [147], in view of the deficiency of existing researches, which is only carried out unilaterally, a combined transaction model of generation rights and emission rights is designed, and the mechanism of combined transaction mode on electricity price, emission volume, and enterprise profit is discussed.

Research has confirmed the positive role of different green trading mechanism combination operation mode for the green development of electric power. With the gradual improvement of the domestic power market and carbon trading market, it is urgent to study the electricity carbon coupling market. Reference [148] pointed out that thermal power, as the current energy to ensure stable power supply, is affected by the national macro policy and market mechanism, so it is urgent to make reasonable compensation for thermal power enterprises to maintain their survival. Based on this, a compensation mechanism for thermal power enterprises with electricity carbon linkage should be proposed. Reference [149] studies the coupling relationship between carbon market and electricity market, analyzes the cost-saving effect of carbon market mechanism and the interaction effect with power market reform and constructs a generation electricity cost model of power market considering carbon emission price. The research results show that the carbon market and the power market have a mutual restriction relationship.

## 6. Countermeasures and Suggestions on Green Development of Electric Power in China

From a perspective of medium and long-term, China is in the rapid development of industrialization and urbanization. In the future, China's economic development will maintain a relatively rapid growth, and energy consumption will continue relatively [150]. From 2020 to 2030, China's thermal power industry is still in a mature development period, and it will transition to a recession period after 2030 [151]. Therefore, in the future, the

thermal power industry will still assume the core power supply of China. China's green power development should focus on the following issues:

1. Strengthen the research on short-term load forecasting and power forecasting technologies for complex power grids with new energy. Improving the short-term load forecasting of power grid with new energy is conductive to formulate the optimal generation, shutdown planning and scheduling scheme, and improving the utilization efficiency of power generation equipment. In addition, it can also enhance the stability and security of the power system [152] and improve the power grid's ability to consume new energy. With the continuous deepening of the electricity market reform, accurate short-term load forecasting and renewable energy power forecasting of complex power grid can provide relevant basis for wholesale sale of grid connected renewable energy under the power market conditions, promote inter-provincial preferential consumption of new energy, and reduce the risk caused by uncertainty of renewable energy for power market participants. At the same time, it can accelerate the spot electricity market of sufficient renewable energy consumption across provinces and regions, promote the consumption of new energy, provide trading support to day-ahead and intraday electricity market, and accelerate the construction of trans provincial power market.

2. Continue to promote the research on clean energy substitution and conversion mechanism, mainly to strengthen the study on the economy, security, and policy of clean energy substitution. Carry out the evaluation of the effect of electric energy substitution conversion, especially the evaluation of energy efficiency, cost, and reliability of crucial energy consuming industries and energy consuming equipment. China should carry out research on industry energy substitution conversion mechanisms based on quota, price, tax rate, and subsidy.

3. Strengthen the exploration of the green development mode of electric power with Chinese characteristics, especially the study of the peak load regulation method of new energy joint PHES. PHES has good economics, superior energy storage capacity, and excellent flexibility and dispatchability. The multi-energy combination mode equipped with PHES can stabilize the fluctuation and flicker of wind and photovoltaic output, reduce the impact of high proportion of new energy grid-connected on the power system, and enhance the operation economy of the system. Meanwhile, strengthen the research and application of multi-energy synergy/combination mode, analyze its complex characteristics in the market environment, so as to participate in the electricity market and carbon market trading as soon as possible, weaken the dominant position of thermal power gradually, enhance the competitiveness of new energy in the market, and accelerate the development process of green power in China.

4. Strengthen the research on the power green trading mechanisms, especially the linkage trading mode of multiple types of green trading mechanisms. For the market with multiple green trading mechanisms, formulate reasonable and efficient initial allocation scheme, pricing method, and management strategy to avoid repeated payment of environmental compensation fees by enterprises, and give full play to the enthusiasm of enterprises for energy conservation and emission reduction to further play the role of market allocation of resources and promote the consumption of new energy, and realize the green transformation of China's power industry. Meanwhile, strengthen publicity and education, popularize knowledge of energy conservation and emission reduction, formulate supporting promotion measures by relevant government departments, enhance public awareness, and promote the implementation of green mechanisms in various industries.

5. There is an urgent need to break down provincial barriers and accelerate the construction of a unified national electricity market with fair competition, health, and order. Due to the reverse distribution of resources and loads in China, the inter-provincial electricity market will become the primary force of new energy consumption in the

future, and the inter-provincial spot market can realize the dynamic balance between the actual power generation capacity and consumption level of new energy, and achieve efficient resource allocation at the national level. This requires the joint efforts of government departments, power grid enterprises, power generation enterprises, and users to break through the constraints of multiple interests, open inter-provincial channels, strengthen top-level planning and design, and speed up the planning and design of cross-regional power grid supporting projects.

6. Accelerate the establishment of a unified national carbon trading market. As a critical breakthrough, the electric power industry will take the lead in launching the carbon emission trading system, which is conducive to cultivating market players, expanding the coverage of the carbon market, and increasing the role of the carbon market in the transformation and upgrading of the energy system. In the future, in addition to the power industry, seven industries will be included in the unified carbon market. It is necessary to strengthen the research on the coordinated operation of carbon market and other green trading mechanisms. The research on price transmission mechanism, coupling mechanism, and optimal management between carbon market and power market will integrate carbon market, carbon trading, quota system, and green power certificate into the reform of power market to make the energy market rich in green power play a decisive role in resource allocation.

7. Strengthen the research on compensation mechanism of auxiliary power service. With the goal of cost, benefit, investment, and other economics, and the requirements of consumption, characteristics, and emissions as constraints, a new peak-shaving auxiliary service market mechanism that promotes large-scale new energy consumption is established as a supporting means for the power market.

## 7. Conclusions

This paper first emphasizes the necessity of accelerating the green development of electric power in China from two aspects of international responsibility and national macroscopic strategy. Based on China's resource endowment characteristics and energy consumption data, this paper points out the current challenges faced by China's green power development. To cope with the increasingly severe shortage of energy resources and environmental pollution, and realize the sustainable development of human society, China is gradually implementing the innovation mode of power green development, and constantly developing and perfecting the system of power green development, implementing the multi-energy synergy/multi-energy combination mode for multi-energy complementary and efficient utilization of various energy, and green development path of electric power in parallel with scientific and reasonable green energy consumption mechanism. Through the above analysis, the following conclusions can be drawn:

1. The multi-energy synergy/multi-energy combination mode is significant for renewable energy systems with wind or photovoltaic participation, which can reduce the cost of power generation, improve the economy and reliability of the system, and reduce the impact of new energy output on the grid. The power green innovation model with multiple coexistence and full use of the complementary characteristics of different energy sources can stabilize the uncertainty of new energy output, assist in the formulation of reliable power generation plans, and assist in the formulation of peak regulation and frequency regulation strategies with better comprehensive benefits. It is beneficial to break through regional constraints and broaden the space for new energy consumption.

2. Scientific and reasonable renewable energy consumption mechanism is the key problem to be solved. As a policy and mechanism tool, green power derivatives can promote the development of renewable energy by market-oriented means. There is a strong correlation between different green trading mechanisms. The effective linkage mode of green trading mechanism can more reasonably calculate the energy cost of

enterprises, reasonably allocate profits, and accelerate the development of electricity greening.

3. For a period of time in the future, China is still in the trend of rapid development of new energy with thermal power as the main support. We should adhere to the innovation mode of green development of electric power and the synergy path of green trading mechanism, and strengthen the research on energy clean alternative conversion mechanism and power auxiliary service compensation mechanism. At the same time, the research on related technologies of complex power grid with new energy should be strengthened, including short-term load and power forecasting technology, and power grid auxiliary peak shaving technology with new energy participation.

**Author Contributions:** Writing—original draft preparation, K.W., M.Y., and J.W.; conceptualization, X.Y. and X.X.; writing—review and editing, D.N. and Y.L. All authors have read and agreed to the published version of the manuscript.

**Funding:** This work is supported by National Key Research and Development Project (Project No. 2020YFB1707800), the 2018 Key Projects of Philosophy and Social Sciences Research, Ministry of Education, China (Project No. 18JZD032), 111 Project, Ministry of Science and Technology of People's Republic of China (Project No. B18021), Natural Science Foundation of China (Project No. 71804045), Natural Science Foundation of Hebei Province, China (Project No. G2020403008).

**Institutional Review Board Statement:** Not applicable. This study did not require ethical approval.

**Informed Consent Statement:** Not applicable. Informed consent was obtained from all subjects involved in the study.

**Data Availability Statement:** Not applicable.

**Conflicts of Interest:** The authors declare no conflict of interest.

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
