# Peer review of "Analysis and Countermeasures of China’s Green Electric Power Development"

_sustainability, doi:10.3390/su13020708_

Round 1

Reviewer 1 Report

This paper analyses countermeasures of China's green electric power development. The key points of the green power development model with the consumption of new energy as the core are reviewed summarizing the key  issues. There is no research performed in the paper, except few descriptive figures and tables. The paper could be classified as professional paper only.  Therefore, I suggest rejection of this paper for publication in journal Sustainability. 

Author Response

Thank you very much for your valuable advice. We are very sorry that this paper did not get your approval.

This paper summarizes the relevant research at home and abroad, and then studies the green development of China's electric power, and the type of the paper is review. Based on the reality of China's power green development, this paper elaborates on the current situation and necessity of electric power green development, combs the existing green development mode of electric power from two aspects of multi-energy synergy and multi-energy combination, and introduces the role of typical multi-energy synergy mode for green development of electric power. Next, this paper compares the significance of various types of multi-energy combination modes for suppressing the uncertainty of new energy output, formulating reliable power generation plan and assisting peak load regulation and frequency regulation. This paper also analyzes China's exploration of green power development innovation, focusing on green derivatives of renewable energy, and summarizes the benefits of different green trading mechanisms in the process of green power development in China, as well as the latest research achievements of collaborative development mode of different types of green trading mechanisms.

We are very sorry that we did not get your affirmation. We always appreciate your contribution to our paper. Sincerely wish you good health and success in your work!

Reviewer 2 Report

The article "Analysis and Countermeasures of China's Green Electric Power Development" provides a very detailed analysis of China's energy system. The topic is topical and important. The directions of transformation and modernization of the energy sector in China were presented. The current situation and the necessity to develop green electricity were discussed. The authors indicated innovative activities in the field of green energy development, focusing on ecological derivatives of renewable energy. The benefits of green trade mechanisms in the process of green energy development in China, as well as the latest research achievements in the development of green trade mechanisms were also indicated.
The structure of the article is correct. The literature review is impressive. All existing and important documents have been considered.
The final conclusions (Chapter 7) are an element that can be improved.
In my opinion, the most important postulates resulting from the article were not taken into account.

Author Response

Dear editor,

Thank you for your letter and for the reviewers’ comments concerning our manuscript. Those comments are all valuable and very helpful for revising and improving our paper, as well as the important guiding significance to our researches. We have studied comments carefully and have made correction which we hope meet with approval. Revised portion are marked in red in the paper. The main revisions are as follows:

We have revised the conclusion carefully to make it clearer and targeted. The revised conclusion can better reflect the research focus of this paper.

The responds to the reviewer’s comments are as follows:

Comment: The article "Analysis and Countermeasures of China's Green Electric Power Development" provides a very detailed analysis of China's energy system. The topic is topical and important. The directions of transformation and modernization of the energy sector in China were presented. The current situation and the necessity to develop green electricity were discussed. The authors indicated innovative activities in the field of green energy development, focusing on ecological derivatives of renewable energy. The benefits of green trade mechanisms in the process of green energy development in China, as well as the latest research achievements in the development of green trade mechanisms were also indicated. The structure of the article is correct. The literature review is impressive. All existing and important documents have been considered. The final conclusions (Chapter 7) are an element that can be improved. In my opinion, the most important postulates resulting from the article were not taken into account.

Response: Thank you very much for your appreciation and valuable suggestion, which is essential for us to improve the quality of our paper. We revised the final conclusion (Chapter 7) to make it clearer and targeted. The revised conclusion can better reflect the research focus of this paper. The revised s conclusions are as follows:

This paper first emphasizes the necessity of accelerating the green development of electric power in china from two aspects of international responsibility and national macroscopic strategy. Based on china's resource endowment characteristics and energy consumption data, this paper points out the current challenges faced by China's green power development. To cope with the increasingly severe shortage of energy resources and environmental pollution, and realize the sustainable development of human society, china is gradually implementing the innovation mode of power green development, and constantly developing and perfecting the system of power green development, implementing the multi-energy synergy/multi-energy combination mode for multi-energy complementary and efficient utilization of various energy, and green development path of electric power in parallel with scientific and reasonable green energy consumption mechanism. Through the above analysis, the following conclusions can be drawn:

  1. The multi-energy synergy/multi-energy combination mode is significant for renewable energy systems with wind or photovoltaic participation, which can reduce the cost of power generation, improve the economy and reliability of the system, and reduce the impact of new energy output on the grid. The power green innovation model with multiple coexistence and full use of the complementary characteristics of different energy sources can stabilize the uncertainty of new energy output, assist in the formulation of reliable power generation plans, and assist in the formulation of peak regulation and frequency regulation strategies with better comprehensive benefits. It is beneficial to break through regional constraints and broaden the space for new energy consumption.
  2. Scientific and reasonable renewable energy consumption mechanism is the key problem to be solved. As a policy and mechanism tool, green power derivatives can promote the development of renewable energy by market-oriented means. There is a strong correlation between different green trading mechanisms. The effective linkage mode of green trading mechanism can more reasonably calculate the energy cost of enterprises, reasonably allocate profits, and accelerate the development of electricity greening.
  3. For a period of time in the future, china is still in the trend of rapid development of new energy with thermal power as the main support. We should adhere to the innovation mode of green development of electric power and the synergy path of green trading mechanism, and strengthen the research on energy clean alternative conversion mechanism and power auxiliary service compensation mechanism. At the same time, the research on related technologies of complex power grid with new energy should be strengthened, including short-term load and power forecasting technology, and power grid auxiliary peak shaving technology with new energy participation.

Thank you for the things you did for our manuscript again! We are looking forwarding to your reply!

Best regards,

Dr. Wang

Round 2

Reviewer 1 Report

This paper could be classified as professional paper, I suggest publication in other specialized journals for this field of research.

This manuscript is a resubmission of an earlier submission. The following is a list of the peer review reports and author responses from that submission.